# A Simplified Method for CRISPR-Cas9 Engineering of *Bacillus subtilis*

Ankita J. Sachla,[a] Alexander J. Alfonso,[a] John D. Helmann[a]

[a]Department of Microbiology, Cornell University, Ithaca, New York, USA

**ABSTRACT** The clustered regularly interspaced short palindromic repeat (CRISPR)-Cas9 system from *Streptococcus pyogenes* has been widely deployed as a tool for bacterial strain construction. Conventional CRISPR-Cas9 editing strategies require design and molecular cloning of an appropriate guide RNA (gRNA) to target genome cleavage and a repair template for introduction of the desired site-specific genome modification. Here, we present a streamlined method that leverages the existing collection of nearly 4,000 *Bacillus subtilis* strains (the BKE collection) with individual genes replaced by an integrated erythromycin (*erm*) resistance cassette. A single plasmid (pAJS23) with a gRNA targeted to *erm* allows cleavage of the genome at any nonessential gene and at sites nearby to many essential genes. This plasmid can be engineered to include a repair template, or the repair template can be cotransformed with the plasmid as either a PCR product or genomic DNA. We demonstrate the utility of this system for generating gene replacements, site-specific mutations, modification of intergenic regions, and introduction of gene-reporter fusions. In sum, this strategy bypasses the need for gRNA design and allows the facile transfer of mutations and genetic constructions with no requirement for intermediate cloning steps.

**IMPORTANCE** *Bacillus subtilis* is a well-characterized Gram-positive model organism and a popular platform for biotechnology. Although many different CRISPR-based genome editing strategies have been developed for *B. subtilis*, they generally involve the design and cloning of a specific guide RNA (gRNA) and repair template for each application. By targeting the *erm* resistance cassette with an anti-*erm* gRNA, genome editing can be directed to any of nearly 4,000 gene disruptants within the existing BKE collection of strains. Repair templates can be engineered as PCR products, or specific alleles and constructions can be transformed as chromosomal DNA, thereby bypassing the need for plasmid construction. The described method is rapid and facilitates a wide range of genome manipulations.

**KEYWORDS** *Bacillus subtilis*, CRISPR-Cas9, genome editing, transformation, allelic replacement, genetics

> "Healing is a matter of time, but it is sometimes also a matter of opportunity."
> – Hippocrates

*Bacillus subtilis* is a Gram-positive model bacterium and serves as an important platform for industry, including the production of commodity chemicals, metabolites, and proteins (1–3). A wide variety of approaches are available for genome modification and editing, which are further aided by the natural competence of many laboratory strains.

One of the earliest methods for introducing directed mutations was transformation with plasmids that lack an active origin in *B. subtilis*. Selection for a plasmid-borne antibiotic resistance gene results in recovery of plasmid integrants, which allows for gene disruptions or the generation of reporter fusions. This approach, as implemented in the pMUTIN series of plasmids (4), was widely employed in early functional genomics studies of *B. subtilis*. However, plasmids integrated by single-crossover can have polar effects, and the integrational event is reversible when selection is removed, leading to genetic instability.

Address correspondence to Ankita J. Sachla, ajs588@cornell.edu, or John D. Helmann, jdh9@cornell.edu.

A simplified method for CRISPR-Cas9 engineering of Bacillus subtilis. New work from Ankita Sacha, Alexander Alfonso, and @johnhelmann introduces a streamlined genome editing method that can bypass time-consuming molecular cloning steps.

Further refinements were soon introduced to allow introduction of mutations that are stable in the absence of ongoing selection. For example, with long-flanking homology PCR (LFH-PCR), double-crossover integration occurs at regions upstream (UP) and downstream (DO) of the desired site of genome modification (5). This can be used to introduce null or altered function mutations, epitope or fluorescent protein tags, or reporter fusions at a specific locus as long as the desired strain can be isolated by selection or identified by screening. If there is no selection, as is often the case, an antibiotic resistance marker can be included in the region between the UP and DO regions of homology for selection. Mutations introduced in this manner are generally stable in the absence of ongoing selection, but engineering strains with multiple mutations can be limited by the available resistance cassettes. This technique was therefore further refined using Cre/Lox-based recombination to remove the antibiotic resistance cassette (6) and by using MazF as a counterselectable marker (7, 8).

The development of the BKE/BKK collection of strains has greatly simplified the generation of *B. subtilis* strains carrying null mutations (9). These collections, with nearly 4,000 strains each, contain erythromycin-resistant (BKE) or kanamycin-resistant (BKK) single-gene replacements and are available from the Bacillus Genetic Stock Center (BGSC, USA) and the National BioResource Project (NBRP, Japan). Typically, null mutations are introduced into the genetic background of choice by transformation of genomic DNA from the chosen BKE or BKK strain with selection for the antibiotic cassette. In a subsequent step, the strain can be by transformed with pDR244, a temperature-sensitive plasmid expressing the Cre recombinase to excise the antibiotic cassette, leaving behind an in-frame gene deletion with a 150-bp (50-amino acid) bar-coded scar (9). This process can be repeated to generate strains containing multiple null mutations.

The CRISPR-Cas9 system, originally from *Streptococcus pyogenes*, has been widely adapted for bacterial genome editing, including several variants for *B. subtilis* (10–13). CRISPR-Cas9 genome editing facilitates genetic manipulations, including the introduction of point mutations, large deletions, epitope or fluorescent protein tags, regulatory site mutations, and expression of nonnative genes. In addition, targeting of inactive nucleases to specific promoter regions allows for interference with gene expression (CRISPRi) and thereby gene repression (10, 14, 15). As often applied, the Cas9 endonuclease is targeted to a specific site by an ∼20-nucleotide (nt) guide RNA (gRNA), and homologous recombination with a cloned repair template allows for chromosome repair and cell survival. The gRNA and repair template can be introduced together on a single-plasmid system as implemented in plasmid pJOE8999 (11). While this plasmid provides a powerful tool for a wide variety of genome manipulations, each construct requires design and cloning of both an appropriate gRNA and the desired repair template containing the intended mutation together with sufficient flanking DNA for homologous recombination. These multiple cloning steps present a bottleneck for the rapid deployment of CRISPR-Cas9 genome engineering. This has motivated efforts to increase the rapidity with which genomes can be edited by using strains with Cas9 expressed from an inducible chromosomal locus (10), plasmids optimized for rapid fragment exchange (16), and introduction of repair templates by cotransformation with PCR products (10) or by recombineering using donor oligonucleotides (17).

Here, we streamline genome editing by leveraging the extant BKE collection of nearly 4,000 *B. subtilis* strains with individual genes replaced by an integrated erythromycin (*erm*) resistance cassette (9). We have designed a plasmid (pAJS23) containing a gRNA targeted to the erythromycin (*erm*) cassette. pAJS23 can be modified to incorporate repair template DNA, or the repair template can be provided by cotransformation with either a PCR product or chromosomal DNA. This approach allows one-step genome editing with no need for gRNA design or plasmid cloning steps. This single plasmid system will allow for greater accessibility and efficiency in genome editing of *B. subtilis*.

## RESULTS

**Plasmid pAJS23 encodes an *erm*-directed gRNA.** Plasmid pJOE8999 has been widely used for CRISPR-Cas9-mediated engineering in *B. subtilis* (11). Rather than

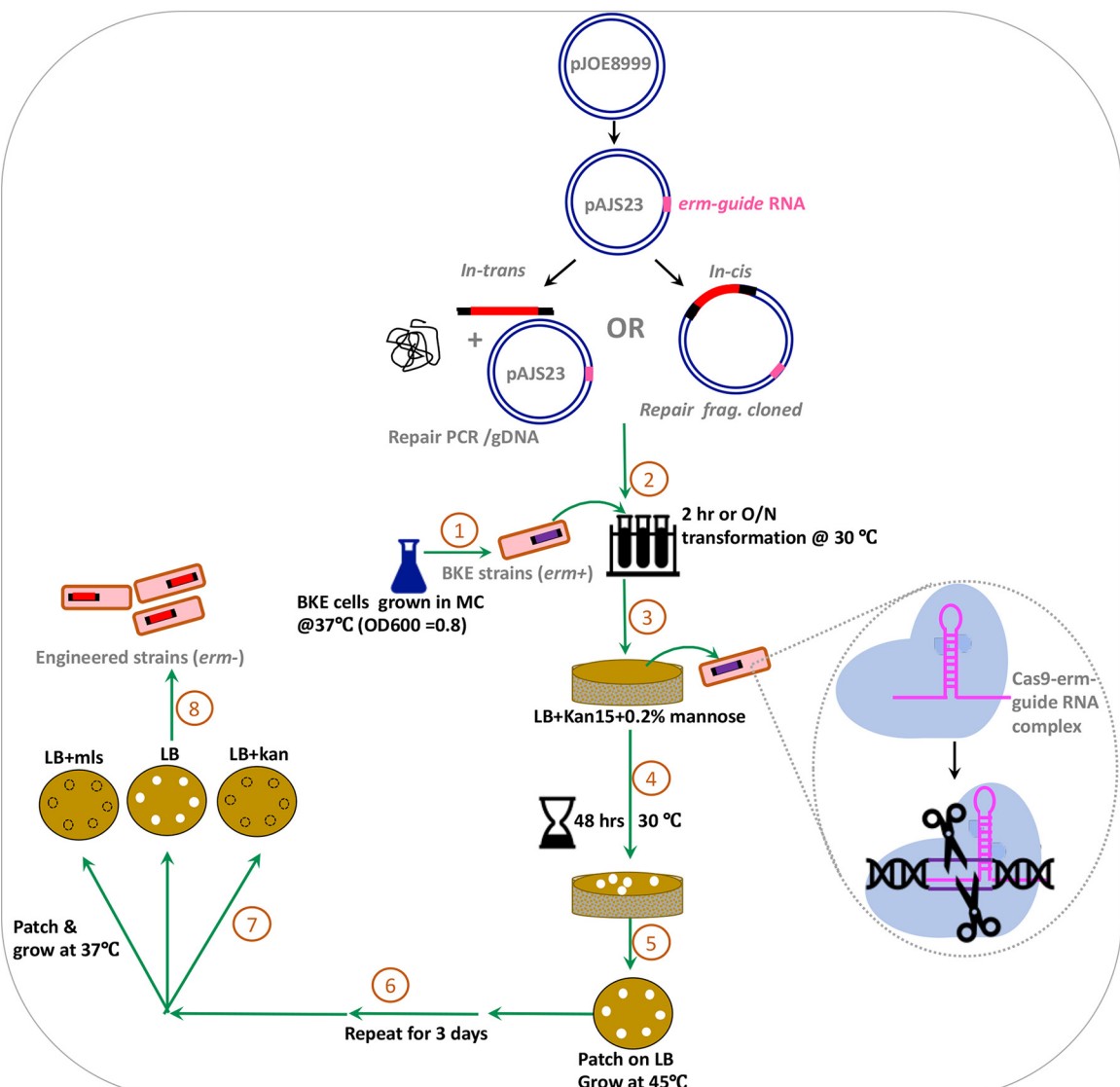

**FIG 1** CRISPR-Cas9 editing with an *erm* gRNA. Plasmid pJOE8999 (7.8 kb; [11]) expresses Cas9 nuclease under a mannose-inducible promoter and is temperature sensitive for replication in *B. subtilis*. Plasmid pAJS23 contains a cloned gRNA directed against the MLS resistance cassette (*erm* gene). This *erm* guide RNA (pink) can be used to target and cleave specific genome regions containing integrated *erm* cassettes as present in the BKE collection of gene disruptants (9). Repair templates can be provided in *trans* (left) by cotransformation of recipient cells with pAJS23 and a repair template (either a PCR product of genomic DNA) or in *cis* (right) by cloning of the repair template into pAJS23. Note that repair templates should include DNA from the regions upstream (UP) and downstream (DO) of the deleted gene in the targeted BKE strain. Engineered strains are cured of the temperature-sensitive plasmid, and MLS^s (*erm*–) colonies are recovered for verification by DNA sequencing. A more detailed explanation is in Fig. S1.

designing a unique gRNA for each application, we sought to leverage the large BKE collection of single-gene *erm* replacements by design of an anti-*erm* gRNA. The designed gRNA is specific to the *erm* gene, and the closest homology in the *B. subtilis* 168 genome has 4 mismatches, which thereby prevents off-target effects. The resultant plasmid, pAJS23, allows selective cleavage of the chromosome at any site with an integrated *erm* gene (Fig. 1). Subsequent to chromosome cleavage, homologous recombination with a cognate repair template restores chromosome integrity to allow survival. Here, we present several examples of genome-editing using

DNA repair templates either cloned into pAJS23, or cotransformed as either a PCR product or genomic DNA.

**Genome editing using pAJS23 with a cloned repair template.** As shown in previous studies, repair DNA templates for CRISPR-Cas-based *B. subtilis* genome editing can be provided by cloning into a plasmid (11) but can also be provided by transformation of PCR products into cells, albeit with reduced efficiency (10). We first explored the utility of pAJS23 for genome editing by cloning appropriate repair templates into pAJS23. For each application, the repair template is generated containing the desired cargo (gene of interest with or without modifications) flanked by UP and DO regions of homology for integration into the chromosome. For cloning of the repair template into pAJS23, the outermost primers used for UP and DO region amplification contain SfiI recognition sites corresponding to the two SfiI sites in the parent plasmid, pJOE8999 (11). Note that SfiI recognizes the sequence GGCCNNNN/NGGCC and leaves a 3-nt 3′ overhang. The two SfiI sites in pJOE8999 leave different overhangs (5′-ACG 3′ and 5′-TAT-3′), and the primers used to amplify the repair template should include SfiI sites that generate 3′ overhangs (5′-CGT-3′ and 5′-ATA-3′) that are compatible with the vector overhangs.

For genome engineering, it is recommended to first transform the relevant gene disruptant from the BKE collection into the desired parent strain prior to genome editing. This helps to ensure that all strains are isogenic except for the modified region. Next, the pAJS23 derivative carrying the desired repair template is transformed into the strain containing an *erm* cassette in the region between the UP and DO regions of homology. Cleavage of the chromosome within the *erm* cassette is a potentially lethal event, and surviving cells are enriched for those where chromosome integrity is restored by recombination at the UP and DO homologous regions.

**Gene replacement at locus.** As a first application, we replaced an endogenous *B. subtilis* gene with an ortholog from another organism. We targeted *cpgA*, the third gene in the *prpC-prkC-cpgA-rpe-yloS* operon. The *cpgA* gene encodes a ribosome assembly GTPase (required for growth at lower temperatures) that additionally functions as a metabolite proofreading phosphatase that is important under glycolytic conditions (18). To determine whether the ribosome-assembly function is conserved among related organisms, we used gene replacement to substitute the *cpgA* coding sequence with that of the *Staphylococcus aureus* ortholog *rsgA*.

For this application, we first generated LFH-PCR products containing three DNA fragments corresponding to (i) an upstream region of homology from *prkC* (UP), (ii) the *Staphylococcus aureus rsgA* gene (*cpgA* ortholog), and (iii) a downstream region of homology from *rpe* (DO). The resulting 2.4-kb LFH-PCR fragment was cloned into pAJS23 between the adjacent SfiI restriction sites to generate pAJS24, which was transformed into BKE15780 (*cpgA::erm trpC2*). Transformants were selected on LB plates at 30°C with kanamycin (to select for the temperature-sensitive plasmid) and mannose (to induce gRNA expression) for 48 h. The resultant Kan$^r$ clones contain the plasmid and express the anti-*erm* gRNA, so there is a strong selection for recombinational repair. We then patched the transformants onto LB plates without antibiotics at 42°C to facilitate plasmid curing. After three passages at 42°C, the majority (89%) of the test clones were MLS sensitive (MLS$^s$), indicative of a loss of the *erm* resistance cassette (Table 1). In each case tested (10/10), the resultant *erm*-sensitive colony contained the desired gene replacement, with *cpgA* replaced by the *S. aureus rsgA* gene (Table 1). These *cpgA::rsgA-Sau* transformants were capable of rescuing growth defects of *cpgA* null mutants as visualized through serial dilution at permissive (37°C) and nonpermissive (30°C) temperature for ribosome assembly defects (Fig. 2A).

**Introduction of an epitope tag.** As a second application, we incorporated a FLAG epitope tag by modification of the *B. subtilis sodA* gene encoding superoxide dismutase. Since *sodA* deletion strains are defective in competence (19), we chose to use the adjacent *yqgC::erm* BKE strain as the recipient. We generated plasmid pAJS25 to include a 3.9-kb repair template containing the desired *sodA*-FLAG gene flanked by upstream (*yqgB-yqgC*) and downstream (*yqgE*) genes (Fig. 2B). In this case, recombinational repair could occur downstream of *sodA*, leading to successful integration of the *sodA*-FLAG gene, or within the *sodA* gene. After curing of the plasmid, most of the recovered clones (66%, Table 1) were MLS$^s$, consistent with loss of the *erm* cassette. All 10 of the MLS$^s$ transformants characterized encoded

**TABLE 1** Summary of efficiency results for *cis* and *trans* repair templates[a]

| Base pairs of homology in repair template (left + right)[b] | Total length of repair template (bp) | Plasmid | No. of Kan[r] colonies | No. of MLS[s] clones | MLS[s] (%)[s] | Efficiency for MLS[s] (%) | Purpose |
|---|---|---|---|---|---|---|---|
| Cloned repair template | | | | | | | |
| 1,534 (750 + 784) | 2,410[c] | pAJS24 | 48 | 43 | 89 | 100 (10/10) | *cpgA::erm→rsgA* (Sau) heterologous complementation |
| 2,599 (1028 + 1571) | 3,893 | pAJS25 | 197 | 131 | 66 | 100 (10/10) | *yqgC::erm→yqgC-sodA*-FLAG epitope fusion |
| 2,188 (725 + 1463) | 4,281[d] | pAJS26 | 53 | 21 | 40 | 100 (15/15) | *yqgC::erm→yqgC-gfp* translation fusion |
| 1,405 (633 + 772) | 2,170 | pAJS27 | 92 | 82 | 89 | 100 (20/20) | *yceF::erm→yceF*(Ile206Thr)[c] |
| 1,719 (1,028 + 691) | 2,202 | pAJS28 | 56 | 46 | 82 | 100 (46/46) | *yqgC::erm→*deletion of intergenic *S936* element |
| 546 (259 + 287) | 696 | pAJS29 | 855 | 186 | 22 | 100 (25/25) | *yqgC::erm→ΔyqgC* (markerless) |
| 1,594 (725 + 869) | 1,744 | pAJS30 | 110 | 97 | 85 | 100 (12/12) | *yqgC::erm→ΔyqgC* (markerless) |
| Cotransformation with repair template[f] | | | | | | | |
| 1,405 PCR (633 + 772) | 2,170 | NA | 63 | 28 | 44 | 18 (5/28)[e] | *yceF::erm→yceF*(Ile206Thr)[c] |
| *yceF*(Ile206Thr)[c] gDNA | NA | NA | 96 | 28 | 27 | 11 (3/28)[e] | *yceF::erm→yceF*(Ile206Thr)[c] |
| *yceF*(Ile206Thr)[c] gDNA | NA | NA | 50 | 38 | 76 | 13 (5/38)[e] | *yceF::erm→yceF*(Ile206Thr)[c] |

[a]NA, not applicable.
[b]Length of UP and DO homology is depicted in parenthesis.
[c]*rsgA* (*cpgA* homolog) of 876 bp was amplified from *S. aureus* Newman strain.
[d]715 bp of GFP coding sequence was amplified from pGFP-star plasmid DNA.
[e]Represents the fact that only 5 out of 28 clones (for PCR-based) and 5 out of 38 clones (for gDNA-based) showed PCR products.
[f]Cotransformation had different size transformants after 2 days on Kan + 0.2% mannose plates, and we found large colonies that appeared rapidly and usually did not show a successful CRISPR event.

the *sodA*-FLAG protein, which was monitored as a function of cell growth by immunoblotting with anti-FLAG rabbit primary antibody and horseradish peroxidase (HRP)-conjugated anti-rabbit mouse secondary antibodies (Fig. 2B).

**Generation of a GFP-translational fusion.** As a third application, we modified the *yqgC* gene to encode a translational green fluorescent protein (GFP) fusion protein. Plasmid pAJS26 was engineered to contain a 4.3-kb repair template encoding *yqgC-gfp* together with flanking upstream and downstream DNA. The *yqgC::erm* strain was the recipient for genome editing. We again recovered MLS[s] colonies, and all those tested had the desired gene fusion as verified by DNA sequencing (Table 1). The resulting strain expressed GFP, as visualized during the late stationary growth with fluorescence microscopy (Fig. 2C).

**Single nucleotide changes.** For a fourth application, we introduced *yceF** (an allele of *yceF* with an Ile206Thr substitution) on a 2.17-kb repair template cloned into pAJS23 to generate pAJS27. The *yceF* gene encodes a membrane protein from the TerC family and is part of the *yceCDEFGH* operon (20). We were again able to recover the desired recombinant strain with high efficiency (Table 1, Fig. 2D).

**Deletion of an intergenic region.** As a fifth example, we deleted the intergenic region within the *yqgC-sodA* operon, which codes for an uncharacterized RNA feature (S936) and includes promoters for transcription of *sodA*. For this construction, the pAJS23 plasmid was modified to include a 2.2-kb repair template resulting in plasmid pAJS28. With *yqgC::erm* as the recipient strain, we recovered the desired genetic construct with high efficiency (Table 1), as confirmed by checking the size of PCR fragments spanning the desired deletion and by DNA sequencing (Fig. 2E).

**Generation of clean deletions.** As a final application, we introduced a *ΔyqgC* deletion derived from the corresponding BKE strain to generate two cloned repair templates differing in the extent of homology to the chromosome. Plasmid pAJS29 had a combined 596 bp of homology for recombination (746 bp total length, including a 150-bp scar region), whereas pAJS30 had 1,594 bp (1,744 bp total length). Both plasmids enabled generation of the desired *ΔyqgC* deletion strain, although pAJS30 gave a higher efficiency (Table 1). In general, for each of these applications, we had no difficulty in identifying MLS[s] clones after plasmid curing, although the frequency of recovery was reduced when the length of the regions of homology for recombination upstream or downstream of the *erm* cassette fell below 300 bp (e.g., pAJS29).

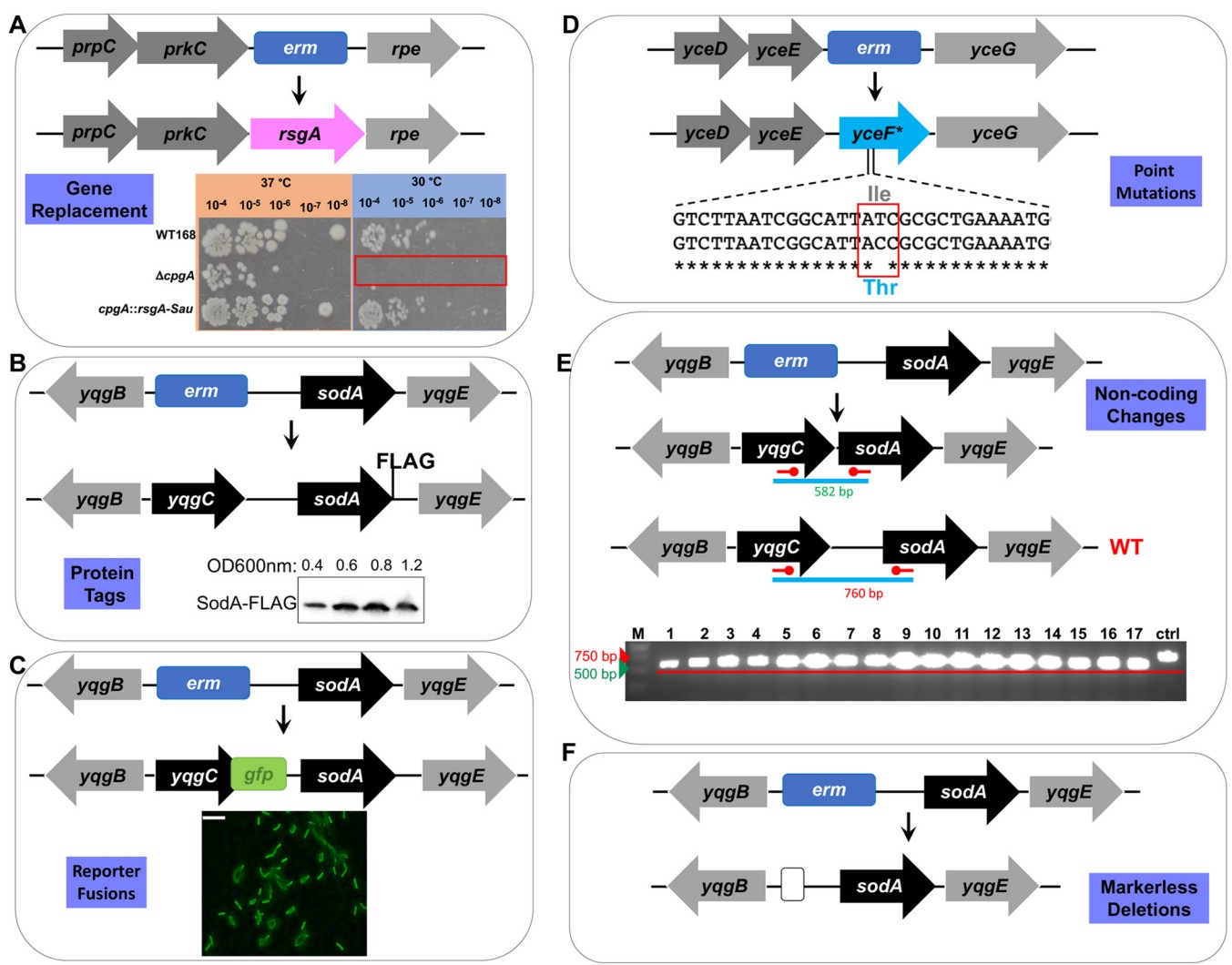

**FIG 2** Representative applications of CRISPR-Cas9 editing of BKE strains. Recipient strains from the BKE collection (with specific genes replaced by an integrated *erm* cassette) were transformed with a pAJS23 derivative carrying a cloned repair template (see Table 1). (A) The *cpgA::erm* recipient was edited to replace the missing *cpgA* gene with the ortholog from *S. aureus* (*Sau-cpgA* = *rsgA*), shown in pink. The *rsgA* gene complements the cold-sensitivity of the *cpgA* mutant strain as shown by serial dilution on LB plates incubated at 30°C (inset). (B) The *yqgC::erm* strain was used to introduce a sequence encoding a FLAG tag (DYKDDDDK) at the C terminus of SodA and simultaneously reintroduce a functional *yqgC* gene. Western blot analysis of cells harvested at the indicated cell density confirms the presence of the 23.5-kDa SodA-FLAG protein (inset). (C) The *yqgC::erm* strain was edited to express a C-terminal YqgC-GFP (green fluorescent protein, *Aequorea victoria*) translational fusion. The resultant strains were confirmed by DNA sequencing, and the expression of GFP was confirmed by fluorescence microscopy of cells grown in LB to the stationary phase (scale bar = 10 $\mu$). (D) The *yceF::erm* strain was edited to express a mutant allele (*yceF**) encoding the YceF Ile206Thr protein. This strain was generated using a cloned repair template (pAJS27; Table 1) or by providing repair template DNA as a PCR product or genomic DNA (Table 1). (E) We used a *yqgC::erm* recipient strain (as shown in panels B and C) to edit the operon by removing the intervening intergenic region, including the RNA feature S936. The desired deletion was confirmed by visualization of correctly sized (582 bp) PCR products (middle inset), which showed a reduction in size relative to the 760-bp control product. (F) A clean deletion of *yqgC*. The *yqgC::erm* locus was replaced by a markerless deletion generated using plasmid pDR244 as described previously (9).

**Provision of a repair template by cotransformation.** Since *B. subtilis* is naturally competent, it is possible to introduce DNA repair templates as a PCR product, as shown previously (10). Therefore, we decided to explore genetic constructions made by cotrans- forming pAJS23 with a repair template provided as either a PCR product or genomic DNA. Since competent cells can take up multiple DNA molecules, those that acquire both the plasmid and the repair template should be enriched among those colonies that pass through the standard selection procedure.

To test this procedure, we used the BKE *yceF::erm* strain as the recipient and introduced *yceF** (Ile206Thr substitution) on a repair template provided either in *cis* (by cloning into pAJS23), as a PCR product, or as genomic DNA (Fig. 2D). When the repair template was pro- vided as a PCR product spanning the *yceF** gene, the efficiency of the procedure decreased

to ~18% (5/28 MLS$^s$ clones were confirmed by sequencing). We were also able to obtain the desired strain with a repair template provided using genomic DNA. In two separate trials using genomic DNA, we observed similar efficiencies (11% and 13%; Table 1). With this strategy, genome editing can be done quickly since there are no required cloning steps, and in many applications the repair template can be provided as genomic DNA. Even though the efficiency in these examples was not as high as with a cloned repair template, it was still straightforward to identify clones with the desired genetic change.

## DISCUSSION

CRISPR-Cas9 and related genome editing systems have revolutionized biology by enabling the precise editing of complex genomes (21, 22). The applications of CRISPR-based tools to bacterial genetics are legion and include the introduction of changes ranging from single nucleotide mutations (11), to large-scale chromosomal deletions (23), the precisely tuned modulation of gene expression using CRISPR-interference based approaches (14, 24), and the targeted integration of large regions of up to 10 kb of DNA using Tn*7*-like integrases (25–27). Because of its importance in molecular biology and numerous anticipated applications in medicine, CRISPR-based laboratories are now included in many undergraduate curricula (28).

Here, we describe an efficient method for engineering the *B. subtilis* 168 genome that bypasses many of the steps that can be rate-limiting during genome engineering. It is relatively easy to move selectable markers from one strain of *B. subtilis* to another by taking advantage of natural competence. As a result, null mutations can be introduced into any desired competent strain using genomic DNA from strains from the BKE collection (9). This collection of nearly 4,000 *erm* gene disruptants is commercially available, and individual strains can be ordered from the *Bacillus* Genetic Stock Center (BGSC). With a desired recipient strain in hand, genome editing can be done by either cloning the desired repair template into pAJS23 or by providing the repair template as a PCR product or genomic DNA from a donor strain. The efficiency of recovery of the desired mutation is generally highest when the repair template is provided on the plasmid. However, the reduced efficiency when using a PCR product or genomic DNA as a repair template may be preferable compared to the additional steps required to clone and verify a plasmid construct for each editing application.

Genomic DNA is taken up very efficiently by competent *B. subtilis* cells. Mutations that confer a selectable phenotype are routinely introduced by transformation. However, moving mutations and genetic constructions that do not confer a selectable phenotype can be more laborious, often involving screening of colonies that have acquired the desired genetic change by congression (29, 30). Using congression, competent cells are selected by transformation with a selectable marker (e.g., antibiotic resistance cassette), and those colonies that have also been transformed with an unlinked mutation are identified by screening. Here, we demonstrate that CRISPR-Cas9-mediated genome cleavage can also provide a strong selection for the transfer of mutations or genetic constructions into a recipient strain, even when it does not confer an immediately selectable phenotype.

One application of this procedure is for studies involving the genetic basis of phenotypes arising during experimental evolution or in laboratory selections and screens for particular phenotypes. The advent of cost-effective whole-genome resequencing (WGS) has made it practical to follow evolutionary changes in great detail as bacteria adapt to different environments. We often use WGS to identify mutations of interest by comparison of an evolved strain to the parent strain. Depending on the nature of the experiment, there may be only one or a few changes detected, or the number may be much larger. It can be challenging to know which of the observed genetic changes is causal for the observed phenotype, and it is always possible that an undetected change is responsible. To be rigorous, it is desirable to reconstruct the evolved strain by introduction of the presumptive causal mutation(s) into the parent strain. This type of genetic backcrossing is important but often nontrivial. Backcrossing of mutations, together with genetic complementation studies, generates assurance that the observed phenotype is correctly linked to a specific genetic change. In previous studies, we

used CRISPR-Cas genome editing with custom gRNAs and repair templates to reconstruct mutations (30–33). For each application, this entailed the design and cloning of an appropriate gRNA, followed by the design and cloning of a repair template. This experience has motivated our current efforts to develop a streamlined procedure that will, in many cases, allow a bypass of all molecular cloning steps and allow strain construction using genetic transformation.

*B. subtilis* is a convenient and safe organism for teaching laboratories. This system is appropriate for use as a simple laboratory exercise where students can modify a genome using CRISPR-Cas9. In addition, many undergraduate laboratories may employ *B. subtilis* for use in authentic research experiences, such as genetic selection or screening of mutants with interesting properties. The underlying genotypic changes can then be identified using WGS-based approaches for a modest cost. CRISPR-Cas9 genome editing provides an effective way to move mutations between strains and thereby test the role of individual mutations in the phenotype of evolved strains.

While the strategy presented here is specific for *B. subtilis* and for the BKE collection of strains, it is straightforward to extend the idea of using a common gRNA to other collections of bacterial strains with indexed gene disruptions (including the related *B. subtilis* BKK collection). Such collections are available in many model organisms, and it is possible in other systems to introduce repair templates as PCR products using electroporation or chemical transformation, as shown for example in *Escherichia coli* (34).

## MATERIALS AND METHODS

**Strains, media, and growth conditions.** All the strains used in this study are listed in Table S1 in the supplemental material, and all the oligonucleotides used in gene amplification, genetic manipulations, and construct verifications are listed in Table S2. All of the *Bacillus* erythromycin knockout (BKE) strains in the 168 background were obtained from BGSC or from the National BioResource Project (NBRP, Japan). Experiments for *E. coli* and *B. subtilis* growth were performed in LB medium (lysogeny broth; 5 ml in a 25-ml tube) or agar (20 ml on a petri plate) with appropriate antibiotics. We used kanamycin in final concentrations of 30 $\mu$g ml$^{-1}$ for *E. coli* and 15 $\mu$g ml$^{-1}$ for *B. subtilis*. The *erm* gene encodes macrolide lincosamide streptogramin (MLS) resistance and is selected with erythromycin (1 $\mu$g ml$^{-1}$) and lincomycin (25 $\mu$g ml$^{-1}$). The 96-well plates with 0.1 ml of LB were inoculated with midlog (optical density [OD] at 600 nm of 0.4) cultures at 50-fold dilutions in final volume for growth measurements. These aerobic cultures were monitored periodically every 60 min for 37°C growth at 600 nm in a Synergy H1 plate reader (BioTek Instruments, Inc., Vermont).

**Erythromycin-specific guide RNA (*erm*-gRNA) cloning.** To target the MLS (*erm*) cassette, we selected the 20-nt sequence 5′TTTGAAATCGGCTCAGGAAA3′, followed by AGG of the protospacer-adjacent motif (PAM) sequence (File S2), which spans the 35th to 41st codon of *erm*. We added BsaI-compatible ends to this sequence and its reverse complement sequence (Table S1), and oligonucleotides were mixed in 1:1 ratio (10 $\mu$M each) in Tris-EDTA (TE) buffer (pH 7.0). This mixture was heated at 95°C for 10 min and cooled to room temperature to generate double-stranded *erm*-gRNA with BsaI-compatible 5′ and 3′ ends (Fig. 1, Fig. S1), prior to cloning into pJOE8999 (11) digested with BsaI. The ligation reaction was transformed into *E. coli* DH5$\alpha$ and plated on LB agar with X-gal (5-bromo-4-chloro-3-indolyl-$\beta$-D-galactopyranoside; 40 $\mu$g ml$^{-1}$) plus kanamycin. White clones (pJOE8999 containing *erm*-gRNA) were selected and subjected to PCR verification and sequencing at the Cornell Biotechnology Resource Center facility to verify the *erm*-gRNA insertion. Culture of *E. coli* harboring *erm*-gRNA in pJOE8999 plasmid (pAJS23) was stored as HEAS25630 (Table S1).

**Genome editing in *Bacillus* with a cloned repair template.** Repair templates were generated using overlap extension (SOEing) PCR or PCR amplification of a suitable region from genomic DNA. The outermost primers were designed to include SfiI restriction sites (see Fig. S1 and S2). The PCR product was digested with SfiI and then ligated into pAJS23 plasmid that had also been digested with SfiI and column-purified to remove the small (12 bp) excised fragment. The repair template and vector were incubated with T4 DNA ligase enzyme at room temperature for 1 h (or 18 h at 16°C) and then transformed into *E. coli* DH5$\alpha$ with selection for Kan$^r$. The desired clone was confirmed with analytical PCR and then transformed into *E. coli* TG1 to generate multimeric plasmid DNA. The resulting pAJS23 derivative could then be used for genome editing of the appropriate strain containing an integrated *erm* cassette.

For each application, we transformed the appropriate recipient strain from the BKE collection with the corresponding pAJS23 derivative. Recipient cells (5 ml) were grown in competence medium to an OD at 600 nm of ~0.8, 1 ml was removed as a negative control, and 1 $\mu$g plasmid was added to the remaining culture and incubated for 2 h at 30°C with shaking (250 rpm). Cells (from 1- ml culture) were plated on LB agar containing 15 $\mu$g ml$^{-1}$ kanamycin (to select for plasmid) and 0.2% mannose (to induce Cas9) and incubated for 24 to 48 h at 30°C. The resulting colonies were passaged three times on LB plates (without any antibiotics) at 45°C for 18 to 24 h each time. Plasmid curing is most efficient when cells are streaked for single colonies at each passage. The resulting clones were then tested for loss of the plasmid by patching on LB plates with MLS, kanamycin, or no antibiotics. Alternatively, colonies may be first patched onto LB plates at 50°C, and then the colony can be patched at 42°C as previously described (11). Clones that failed to grow when patched onto both MLS and kanamycin test plates were selected. The presence of the desired genome change was confirmed using PCR amplification and Sanger sequencing.

**pAJS24 (*rsgA*).** We amplified three overlapping fragments by PCR—*Bacillus prkC* fragment (767 bp), the *rsgA* gene (*cpgA* ortholog) from *S. aureus* Newman (876 bp), and *Bacillus rpe* (799 bp). These PCR fragments were fused to generate a single PCR fragment consisting of SfiI restriction sites for cloning into pAJS23.

**pAJS25 (*sodA*-FLAG).** For this FLAG epitope fusion, we chose LFH PCR-based in-frame *sodA* open reading frame (ORF) fusion to nucleotides GATTATAAAGATGATGATGATAAA coding for an extended DYKDDDDK fusion tag on Mn-SOD protein.

**pAJS26 (YqgC-GFP).** For *yqgC* gene fusion to *gfp*, we generated chimeric PCR by individually amplifying three overlapping fragments. One fragment consisted of *yqgB-yqgC* (1,205 bp), and the second fragment consisted of 712-bp long *yqgC* adaptors-*gfp* (taken from pGFP-star plasmid [35])-*sodA* adaptors, and *sodA-yqgE* fragment (1,467 bp).

**pAJS27 (*yceF**\*).** To introduce a point mutation in *yceF*, we employed LFH primers encoding an Ile206Thr in *yceF* (*yceF**\*). In this approach, we generated and joined two fragments into a single fragment.

**pAJS28 (ΔS936).** To delete the S936 RNA feature in the noncoding region between the *yqgC-sodA* elements, we engineered a 177-bp deletion spanning reference positions from 2586043 to 2586220 in the *Bacillus* 168 genome (GenBank accession number NC_000964.3). We ordered a gBlock with SfiI recognition sequences at both ends from IDT (Integrated DNA Technologies, Inc., Iowa, USA), which was cloned into pAJS23 to generate pAJS25.

**pAJS29 and pAJS30 (Δ*yqgC*).** To convert *yqgC*::*erm* to Δ*yqgC*, we selected primers such that the length of the repair template was either 546 bp or 1,594 bp (Table 1). These repair templates were PCR amplified from Δ*yqgC* genomic DNA (generated from *yqgC*::*erm* using pDR244 as previously described [9]). The resultant strain had an unmarked *yqgC* deletion (Δ*yqgC*) with a 150-bp scar in place of the *erm* resistance cassette.

**Genome editing by cotransformation with a repair template.** A total of 6 ml of recipient *Bacillus subtilis* strain was transformed as described above with 1 $\mu$g of pAJS23 plasmid DNA mixed with either 1 to 2 $\mu$g of PCR DNA or 1 $\mu$g of sheared/sonicated (20 s, 40% amplitude) genomic DNA. From this 6 ml, 1 ml of culture was removed as a negative control. The 5 ml competent cells were incubated with repair template DNA at 30°C overnight and then split into 5 × 1-ml aliquots. Cells were centrifuged for 3 min at 5,000 rpm, resuspended in sterile phosphate-buffered saline (PBS)/LB (0.1 ml), and plated onto LB containing kanamycin and 0.2% mannose for 36 to 48 h at 30°C. Plasmids were cured at 42°C, and the recovered clones were confirmed by Sanger sequencing as described above.

**Spot dilution assay.** The assay was performed as previously described (18). Briefly, cultures were streaked onto an LB agar plate and incubated at 37°C for growth. These cultures were then grown in liquid LB broth (5 ml) until the OD at 600 nm was 0.4. Then serial dilution was performed for cells in 96-well microtiter plates. Next, 10 $\mu$l of serially diluted cells were spotted onto LB agar and allowed to aseptically air-dry. These plates were then incubated at either 37 or 30°C. Plates were imaged after 18 h.

**Western blotting.** The *sodA*-FLAG-tagged HBYL1249 cells were grown in LB at 37°C, and 0.5 ml of cell culture was harvested at 0.4, 0.6, 0.8, and 1.0 OD at 600 nm by centrifugation at 13,000 rpm for 5 min. Cell pellets were suspended with 50 $\mu$l of PBS and 1.0 $\mu$g of lysozyme and incubated at 37°C for 20 min. This culture lysate was centrifuged at 13,000 rpm for 1 min, soluble fraction was collected, and protein concentration was determined by Bradford reagent (Bio-Rad) and separated by SDS-PAGE. Proteins were transferred to polyvinylidene difluoride (PVDF) membrane using a Trans-Blot Turbo transfer system (Bio-Rad). The membrane was blocked (for 1 h) and incubated for 1 h with primary rabbit anti-FLAG antibody (Sigma-Aldrich). Following buffer wash, the membrane was incubated for 1 h with HRP-conjugated IgG mouse anti-rabbit antibodies (Thermo Fisher Scientific). The blot was stained with 1:1 enhancer:substrate reagents (Bio-Rad) for 5 min and visualized with a ChemiDoc MP imaging system (Bio-Rad) and ImageLab software.

**Epifluorescence microscopy.** Candidate overnight-grown cells were collected from the plate and mixed in LB to get 0.1 OD. These cells were spotted onto an agarose pad and gently covered with a coverslip and observed using an Olympus BX61 epifluorescence (for GFP) microscope, and images were captured.

## SUPPLEMENTAL MATERIAL

Supplemental material is available online only.

**SUPPLEMENTAL FILE 1**, PDF file, 1.2 MB.

## ACKNOWLEDGMENTS

We thank the National Institutes of Health for grant R35GM122461 awarded to J.D.H.

We are grateful to Josef Altenbuchner for creating pJOE8999, a pioneering plasmid for *Bacillus* genome editing.

We declare no conflicts of interest.

A.J.S. performed all the experiments, J.D.H. and A.J.S. conceptualized and designed the experiments and analyzed the data, and A.J.S. and A.J.A. drafted the manuscript. J.D.H. supervised, edited, and finalized the manuscript.

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
