## [Reviewer comments · Microbiology Spectrum]

Microbiology Spectrum

A simplified method for CRISPR-Cas9 engineering of *Bacillus subtilis*

Ankita Sachla, Alexander Alfonso, and John Helmann

Corresponding Author(s): John Helmann, Cornell University

Review Timeline:

Submission Date:	June 30, 2021
Editorial Decision:	August 2, 2021
Revision Received:	August 4, 2021
Accepted:	August 16, 2021

Editor: Tino Polen

Reviewer(s): Disclosure of reviewer identity is with reference to reviewer comments included in decision letter(s). The following individuals involved in review of your submission have agreed to reveal their identity: Sang Jun Lee (Reviewer #3)

Transaction Report:

DOI: <https://doi.org/10.1128/Spectrum.00754-21>

August 2, 2021

Prof. John D Helmann
Cornell University
Department of Microbiology
Wing Hall
Cornell University
Ithaca, NY 14853-8101

Re: Spectrum00754-21 (A simplified method for CRISPR-Cas9 engineering of *Bacillus subtilis*)

Dear Professor John D. Helmann,

I have received three reviews of your manuscript from three experts. You will see from the comments below that all believe that your study and manuscript are valuable and provide helpful information for *B. subtilis* genome editing and the systematic analysis of its genetics, physiology, and metabolism, yet the manuscript requires some modifications. Among other points, the experts suggested to address the low frequency of correct clones when co-transforming DNA and add more detailed explanations of your methods to improve the accessibility for non-experts. You should also address the recent advances of CRISPR/Cas technologies in the engineering of microbial genomes, and describe pros and cons of your new system, compared to other studies of CRISPR/Cas microbial genome engineering. Some parts of your manuscript should be re-written for better clarity. You should also consider already in the title, that your method is applicable for the BKE collection and not *B. subtilis* in general. I hope you find the reviewers' comments helpful and can revise your manuscript accordingly, and look forward to receiving a revised version from you.

Thank you for submitting your manuscript to Microbiology Spectrum.

When submitting the revised version of your paper, please provide (1) point-by-point responses to the issues raised by the reviewers as file type "Response to Reviewers," not in your cover letter, and (2) a PDF file that indicates the changes from the original submission (by highlighting or underlining the changes) as file type "Marked Up Manuscript - For Review Only". Please use this link to submit your revised manuscript - we strongly recommend that you submit your paper within the next 60 days or reach out to me. Detailed information on submitting your revised paper are below.

Link Not Available

Sincerely,

Dr. Tino Polen
Editor, Microbiology Spectrum

Journals Department
Reviewer comments:

Reviewer #1 (Comments for the Author):

Dear authors,
the manuscript entitled „A simplified method for CRISPR-Cas9 engineering of *Bacillus subtilis*“ is a manuscript with general interest for the scientific community working with *B. subtilis*. In general, it is well written but can be rephrased in some parts to make it straighter. I think the title is a bit misleading, since the approach is not suitable for the wild type strain, thus it is only applicable for the BKE collection, what should be mentioned in the title. Targeting the *erm*-gene is quite obvious for this collection. The manuscript describes different modifications which were performed, and it is sometimes hard to follow them, and to see what the outcome is. I would suggest to rewrite the description of the different experiments to make it clearer what was done and what is really the novel approach. In general, the manuscript is written well but very often *sought* is used, what might be replaced. Also the efficiency by using gDNA or PCR fragments is quite low compared to the cloned HR repair templates, this should be discussed a bit more.

Minor:

- In table 1 there is missing a space in-between left + right
- What is the reason for the different length of the US and DS fragments?
- Lane 220 is it really high yield, or is it more efficiency?
- Lane 235 how can a construct itself be successful? Would rewrite that insuccessfully deleted or so on...
- Lane 417, as described in 817), would use the author name here.

Reviewer #2 (Comments for the Author):

Summary

Sachla et al. describe a simplified method to modify the *Bacillus subtilis* genome using CRISPR-Cas9 editing and a pre-existing collection of *B. subtilis* strains with individual genes replaced by erythromycin (*erm*) resistance cassette (BKE collection). They design a guide RNA that targets the *erm* cassette and allows Cas9-mediated cleavage at the *erm* gene in any of the BKE knockout strains, and through homologous recombination a DNA repair template modifies the genome at the cleavage site. They demonstrate the application of this method to several genome-editing examples. While CRISPR-Cas9 editing was already established in *B. subtilis*, this method removes the guide RNA design step by only requiring the *erm* guide established in this paper. The editing plasmid and methods presented here are a valuable resource for the *B. subtilis* field that will speed strain construction and improve flexibility in introducing mutations.

General comments

1. The % efficiency for MLSS is 100% for all "cloned repair template" results but is very low for "co-transformation with repair template" (Table 1). This high failure rate by co-transformation is surprising, but the authors do not address the cause of this issue. Why is there a large fraction of MLSS clones that do not have the expected genotype? Is it possible that larger deletions flanking the erm gene occurred, or is there another explanation? I think other researchers may be reluctant to attempt the co-transformation approach if they aren't sure of the outcomes-the authors should try to address this by genotyping a few of the unexpected MLSS mutants.
2. The paper assumes that the audience has extensive knowledge of *B. subtilis* and has a solid understanding of CRISPR-Cas9 experiments. I appreciate the workflow detail in Supp Fig S1; however the authors should also write out each step in the S1 figure legend to further clarify what is happening at each step, particularly for readers with less background knowledge on *B. sub* and CRISPR. Additionally, when they mention Fig 1 in the text, they should also reference that a more detailed explanation of the process can be found in S1.

Specific comments

1. In lines 180-186, the way the text is written suggests that 42{degree sign}C growth causes the loss of erm resistance; however this actually causes the loss of plasmid (kan resistance). It would be helpful to break these steps apart to make it clear that the loss of erm resistance comes from CRISPR repair at 30{degree sign}C, with a separate sentence explaining loss of plasmid (kan) comes from growth at 42{degree sign}C. Additionally, the efficiency of loss of the plasmid is never mentioned but is important information to include.
2. In line 287 the authors briefly mention genetic congression, but their explanation was incomplete. The authors should mention how congression could affect the outcomes of the experiment, such as with using gDNA as a repair template.
3. In line 143, the title "Genome-editing using pAJS23 with a cloned guide RNA" should probably read "a cloned repair template" instead.
4. There may be a typo in the second to last row of Table 1, the calculation for % efficiency for MLSS used 3/28 but the previous column indicated that 26 colonies were tested.

Reviewer #3 (Comments for the Author):

A simplified method for CRISPR-Cas9 engineering of *Bacillus subtilis*

Authors (Sachla AJ et al.) would like to report a simple method for genetic manipulation of *Bacillus subtilis*. Authors designed a CRISPR/Cas9 system that recognizes the erythromycin cassette gene commonly found in the individual gene KO collection of *Bacillus subtilis*. Authors carried out gene replacement between erythromycin resistance cassette and various designed DNAs (plasmids, PCR products etc.) for FLAG-tag fusion, single point mutation, non-coding region changes, GFP fusion, markerless deletion. This study will be helpful in the systematic analysis of genetics, physiology, and metabolism of *Bacillus subtilis*.

However, authors need to address the recent advances of CRISPR/Cas technologies in the engineering of microbial genomes, and to describe pros and cons of authors' new system, compared to other studies of CRISPR/Cas microbial genome engineering.

Following points need to be addressed.

Line 41, "with no requirement for intermediate cloning steps" can be deleted because "this strategy

bypasses the need for gRNA design" is already stated in the same sentence.

Line 52, "free of off-target effects" can be deleted because genomic complexity is not high in the microbial genome. Authors don't need to emphasize "free of off-target effects" in the microbial system.

Line 265 Single nucleotide genome editing can be achieved by oligo-directed mutagenesis followed by negative selection of CRISPR/Cas9 or Cpf1. Authors need to refer to one of the references (PMID 34261850, 34208669, 32807756, 32327447).

Staff Comments:

Preparing Revision Guidelines

For complete guidelines on revision requirements, please see the Instructions to Authors at [link to page]. **Submissions of a paper that does not conform to Microbiology Spectrum guidelines will delay acceptance of your manuscript.**

Please return the manuscript within 60 days; if you cannot complete the modification within this time period, please contact me. If you do not wish to modify the manuscript and prefer to submit it to another journal, please notify me of your decision immediately so that the manuscript may be formally withdrawn from consideration by Microbiology Spectrum.

If you would like to submit an image for consideration as the Featured Image for an issue, please contact Spectrum staff.

Corresponding authors may join or renew ASM membership to obtain discounts on publication fees. Need to upgrade your membership level? Please contact Customer Service at

Service@asmusa.org.

Reviewer #1 (Comments for the Author):

Dear authors, the manuscript entitled „A simplified method for CRISPR-Cas9 engineering of *Bacillus subtilis*“ is a manuscript with general interest for the scientific community working with *B. subtilis*. In general, it is well written but can be rephrased in some parts to make it straighter.

**** Thank you for your supportive comments. We have edited the text for clarity as noted below and as shown in the “marked” version that is uploaded.**

I think the title is a bit misleading, since the approach is not suitable for the wild type strain, thus it is only applicable for the BKE collection, what should be mentioned in the title. Targeting the *erm*-gene is quite obvious for this collection.

**** We considered modifying the title, but decided against it. Very few scientists, even in the *B. subtilis* community, work directly in the “BKE” collection of strains, usually because each lab has its own wild-type (parent) strain. As a first step, it is always best to move *erm* mutations in your favorite gene (i.e. *yfg::erm*) into your own lab’s wild-type strain. This might be a version of the type strain *B. subtilis* 168, but could also be any of a number of related strains. Historically, scientists from the Losick lab use “PY79”, from the Hoch lab “JH642” and those studying biofilms and motility may use the “wild” isolate *B. subtilis* 3610 (close relative to the original Marburg strain). These strains have a common core genome, but differ in their content of phages and mobile elements. *All of these strains can be used as recipients for mutations from the BKE collection.***

In addition, the genetic tools (plasmids, vectors, drug cassettes) routinely used in *B. subtilis* (including *erm*) are often used for studies in other Bacilli. We presented this new approach to the Bacillus community at the recent on-line “Subtillery” conference and this generated a lot of interest. We were contacted by Dr. Filho (Brazil) who wishes to apply this approach to *Bacillus velezensis* FZB (formerly *B. amyloliquefaciens* FZB). It can also be applied to members of the *B. cereus* group (*B. thuringiensis*, *B. anthracis*).

The manuscript describes different modifications which were performed, and it is sometimes hard to follow them, and to see what the outcome is. I would suggest to rewrite the description of the different experiments to make it clearer what was done and what is really the novel approach.

**** Figure 2 presents readers with a schematic that shows the starting genome organization, and how this method can introduce a variety of modifications using selection against the *erm* cassette. Together with the description in the text, we think this is quite clear.**

The novelty of the approach is not the type of constructions that can be made, but the idea of using a single sgRNA to target any of the nearly 4000 non-essential genes (and essential genes with a nearby *erm* cassette) in one of the most important model organisms (*B. subtilis*). This completely bypasses the need to design, synthesize, and clone the DNA encoding the sgRNA. Combined with the option of using genomic DNA as repair template, this greatly simplifies and speeds CRISPR-based editing.

In general, the manuscript is written well but very often *sought* is used, what might be replaced.

****We thank reviewer for this point, and we noted five occurrences of the word “sought.” We have removed four of these.**

Also the efficiency by using gDNA or PCR fragments is quite low compared to the cloned HR repair templates, this should be discussed a bit more.

**** We discuss this at the end of the results section and in the discussion. Even though the efficiency in these examples was not as high as with a cloned repair template, it was still straightforward to identify clones with the desired genetic change. Many procedures in genetic engineering do not proceed with high efficiency. This is why “blue-white” selections were developed for identifying plasmid inserts, for example.**

Minor:

- In table 1 there is missing a space in-between left + right

**** corrected**

- What is the reason for the different length of the US and DS fragments?

**** there is no particular reason. These types of differences are due to the position of the primers used for PCR. Ultimately, this also illustrates the broad range of repair template size that can be used.**

- Lane 220 is it really high yield, or is it more efficiency?

**** corrected**

- Lane 235 how can a construct itself be successful? Would rewrite that insuccessfully deleted or so on...

**** corrected**

- Lane 417, as described in 817), would use the author name here.

**** corrected. Since this is a paper by Sachla and Helmann, we changed to “as previously described (17).”**

Reviewer #2 (Comments for the Author):

Summary

Sachla et al. describe a simplified method to modify the *Bacillus subtilis* genome using CRISPR-Cas9 editing and a pre-existing collection of *B. subtilis* strains with individual genes replaced by erythromycin (*erm*) resistance cassette (BKE collection). They design a guide RNA that targets the *erm* cassette and allows Cas9-mediated cleavage at the *erm* gene in any of the BKE knockout strains, and through homologous recombination a DNA repair template modifies the genome at the cleavage site. They demonstrate the application of this method to several genome-editing examples. While CRISPR-Cas9 editing was already established in *B. subtilis*, this method removes the guide RNA design step by only requiring the *erm* guide established in this paper. The editing plasmid and methods presented here are a valuable resource for the *B. subtilis* field that will speed strain construction and improve flexibility in introducing mutations.

**** We thank the referee for these supportive comments.**

General comments

1. The % efficiency for MLSS is 100% for all "cloned repair template" results but is very low for "co-transformation with repair template" (Table 1). This high failure rate by co-transformation is surprising, but the authors do not address the cause of this issue. Why is there a large fraction of MLSS clones that do not have the expected genotype? Is it possible that larger deletions flanking the *erm* gene occurred, or is there another explanation? I think other researchers may be reluctant

to attempt the co-transformation approach if they aren't sure of the outcomes-the authors should try to address this by genotyping a few of the unexpected MLSS mutants.

**** Many procedures in molecular genetics occur with low frequency, but this does mean they are not useful. This is typical of molecular cloning experiments, where scientists routinely screen for correct clones by PCR (or blue-white selection). We have thought about why this efficiency is low, but do not have a detailed mechanistic answer. We think it unlikely to be due to NHEJ, since this is low frequency in *B. subtilis* and we did not detect deletions in the *erm* gene by PCR (as might be expected if there was NHEJ). We hope that as this technique finds wider use scientists will develop ways of incrementally improving the efficiency. However, having to screen a dozen colonies to find one or two that are correct is not a big barrier to adoption of this technique, and if this fails it is always possible to use cloned repair templates.**

2. The paper assumes that the audience has extensive knowledge of *B. subtilis* and has a solid understanding of CRISPR-Cas9 experiments. I appreciate the workflow detail in Supp Fig S1; however the authors should also write out each step in the S1 figure legend to further clarify what is happening at each step, particularly for readers with less background knowledge on *B. sub* and CRISPR. Additionally, when they mention Fig 1 in the text, they should also reference that a more detailed explanation of the process can be found in S1.

**** We have added additional details in an expanded Fig. S1 legend. We have added references to Fig. S1 in the main text where appropriate.**

Specific comments

1. In lines 180-186, the way the text is written suggests that 42{degree sign}C growth causes the loss of *erm* resistance; however this actually causes the loss of plasmid (kan resistance). It would be helpful to break these steps apart to make it clear that the loss of *erm* resistance comes from CRISPR repair at 30{degree sign}C, with a separate sentence explaining loss of plasmid (kan) comes from growth at 42{degree sign}C. Additionally, the efficiency of loss of the plasmid is never mentioned but is important information to include.

**** The text states, "We then patched the transformants onto LB plates without antibiotics at 42 °C to facilitate plasmid curing." The efficiency of plasmid curing can be quite high, but this depends on the number of passages at elevated temperature and whether investigators simply "patch" or streak for single colonies at each passage. This is clarified in the methods description.**

2. In line 287 the authors briefly mention genetic congression, but their explanation was incomplete. The authors should mention how congression could affect the outcomes of the experiment, such as with using gDNA as a repair template.

**** The description of congression has been clarified.**

3. In line 143, the title "Genome-editing using pAJS23 with a cloned guide RNA" should probably read "a cloned repair template" instead.

**** Yes, thank you for catching that. It is now corrected.**

4. There may be a typo in the second to last row of Table 1, the calculation for % efficiency for MLSS used 3/28 but the previous column indicated that 26 colonies were tested.

**** This is now corrected**

Reviewer #3 (Comments for the Author):

A simplified method for CRISPR-Cas9 engineering of *Bacillus subtilis*

Authors (Sachla AJ et al.) would like to report a simple method for genetic manipulation of *Bacillus subtilis*. Authors designed a CRISPR/Cas9 system that recognizes the erythromycin cassette gene commonly found in the individual gene KO collection of *Bacillus subtilis*. Authors carried out gene replacement between erythromycin resistance cassette and various designed DNAs (plasmids, PCR products etc.) for FLAG-tag fusion, single point mutation, non-coding region changes, GFP fusion, markerless deletion. This study will be helpful in the systematic analysis of genetics, physiology, and metabolism of *Bacillus subtilis*.

**** We thank the referee for these supportive comments.**

However, authors need to address the recent advances of CRISPR/Cas technologies in the engineering of microbial genomes, and to describe pros and cons of authors' new system, compared to other studies of CRISPR/Cas microbial genome engineering.

**** We have cited previously application commonly used in *B. subtilis*, and have added the suggested references to oligo-based methods cited below. A full description of all the many ways that CRISPR/Cas has been used for microbial genome engineering is best left to a review article.**

Following points need to be addressed.

Line 41, "with no requirement for intermediate cloning steps" can be deleted because "this strategy bypasses the need for gRNA design" is already stated in the same sentence.

**** The reference to cloning steps refers to the ligation of the DNA encoding the gRNA into the plasmid. With many (but not all) CRISPR-based systems it is also necessary to clone a repair template into the plasmid. These are the intermediate cloning steps. The first part of the sentence refers to design of the gRNA, which is separate and precedes cloning.**

Line 52, "free of off-target effects" can be deleted because genomic complexity is not high in the microbial genome. Authors don't need to emphasize "free of off-target effects" in the microbial system.

**** deleted as suggested**

Line 265 Single nucleotide genome editing can be achieved by oligo-directed mutagenesis followed by negative selection of CRISPR/Cas9 or Cpf1. Authors need to refer to one of the references (PMID 34261850, 34208669, 32807756, 32327447).

**** We agree that the review article mentioned (34261850) is a very good recent review (which appeared after drafting of our paper). We have now cited this in the Introduction. These other papers (also from Sang Jun Lee's group) are more focused on the use of oligonucleotides as a mechanism to provide a repair template during CRISPR mutagenesis. This is quite distinct from the dsDNA repair templates we use, and discussion of this different approach is not relevant for the studies we report.**

August 16, 2021

Prof. John D Helmann
Cornell University
Department of Microbiology
Wing Hall
Cornell University
Ithaca, NY 14853-8101

Re: Spectrum00754-21R1 (A simplified method for CRISPR-Cas9 engineering of *Bacillus subtilis*)

Dear Professor John D. Helmann,

thank you for your revised version.

It is a pleasure to accept your manuscript in its current form for publication in Microbiology Spectrum.

I am forwarding it to the ASM Journals Department for publication. You will be notified when your proofs are ready to be viewed.

Sincerely,

Dr. Tino Polen
Editor, Microbiology Spectrum
